# From Rheumatoid Factor to Anti-Citrullinated Protein Antibodies and Anti-Carbamylated Protein Antibodies for Diagnosis and Prognosis Prediction in Patients with Rheumatoid Arthritis

**DOI:** 10.3390/ijms22020686

**Published:** 2021-01-12

**Authors:** Chao-Yi Wu, Huang-Yu Yang, Shue-Fen Luo, Jenn-Haung Lai

**Affiliations:** 1Division of Allergy, Asthma, and Rheumatology, Department of Pediatrics, Chang Gung Memorial Hospital, Taoyuan 33303, Taiwan; joywucgu@hotmail.com; 2College of Medicine, Chang Gung University, Taoyuan 333, Taiwan; hyyang01@gmail.com; 3Department of Nephrology, Chang Gung Memorial Hospital, Taoyuan 333, Taiwan; 4Division of Allergy, Immunology, and Rheumatology, Department of Internal Medicine, Chang Gung Memorial Hospital, Chang Gung University, Taoyuan 333, Taiwan; lsf00076@adm.cgmh.org.tw; 5Graduate Institute of Medical Science, National Defense Medical Center, Taipei 114, Taiwan

**Keywords:** rheumatoid factors, anti-citrullinated protein antibodies, anti-carbamylated protein antibodies, rheumatoid arthritis

## Abstract

Rheumatoid arthritis (RA) is a chronic systemic inflammatory disease mainly involving synovial inflammation and articular bone destruction. RA is a heterogeneous disease with diverse clinical presentations, prognoses and therapeutic responses. Following the first discovery of rheumatoid factors (RFs) 80 years ago, the identification of both anti-citrullinated protein antibodies (ACPAs) and anti-carbamylated protein antibodies (anti-CarP Abs) has greatly facilitated approaches toward RA, especially in the fields of early diagnosis and prognosis prediction of the disease. Although these antibodies share many common features and can function synergistically to promote disease progression, they differ mechanistically and have unique clinical relevance. Specifically, these three RA associating auto-antibodies (autoAbs) all precede the development of RA by years. However, while the current evidence suggests a synergic effect of RF and ACPA in predicting the development of RA and an erosive phenotype, controversies exist regarding the additive value of anti-CarP Abs. In the present review, we critically summarize the characteristics of these autoantibodies and focus on their distinct clinical applications in the early identification, clinical manifestations and prognosis prediction of RA. With the advancement of treatment options in the era of biologics, we also discuss the relevance of these autoantibodies in association with RA patient response to therapy.

## 1. Introduction: Overview of Autoantibodies in Rheumatoid Arthritis

Rheumatoid arthritis (RA) is a systemic inflammatory disorder that mainly involves articular inflammation. Approximately 0.5% to 1% of the population worldwide is affected by the disease, which may result in joint destruction and disability [1]. In addition, approximately 40% of patients present with extra-articular manifestations, complicating disease progression and mortality [2]. Autoantibodies isolated from patient serum and synovial fluid play critical roles in the pathogenesis of RA [3]. According to the literature, 70–80% of patients with RA are positive for autoantibodies (autoAbs), such as rheumatoid factors (RFs) and anti-citrullinated protein antibodies (ACPAs) [4]. Although they are not necessarily present in all patients, antibodies reacting with self-antigens, including immunoglobulins and posttranslational modified (PTM) protein epitopes, have been known for over 80 years to exist in RA [3]. With the advancement of our knowledge on the correlation of autoAbs and RA, these antibodies have become critical biomarkers widely utilized in clinical practice for disease prediction and diagnosis and may assist in choosing therapeutic regimens, as depicted in Figure 1.

RFs were the first autoAb reported in RA. They were described by Waaler in 1940 as factors with hemagglutinating activity in the serum of a patient with RA [5] and named by Pike in 1949 for their associations with RA [6]. Subsequent studies have revealed that the presence of RF is associated with a more severe and erosive RA phenotype. Compellingly, utilizing an analytical ultracentrifuge technique, Kunkel and his colleagues later discovered that RF is an antibody to antigen–antibody complexes [7,8]. Moreover, scientists discovered that RF targets antigenic epitopes within the crystallizable fragment (Fc) region of immunoglobulin G (IgG) and is present in various isotypes [9]. Although the specificity of RF for RA is only approximately 60–70% and the conditions required to break tolerance to IgG are not yet fully understood [10], RF was included in the 1987 ACR classification criteria for RA [11] and considered the paradigm of autoAb clinical significance in RA.

The identification of ACPAs can be traced back to 1964, when Nienhuis and Mandema reported anti-perinuclear factors within the sera of RA patients [12]. Approximately 30 years later, Hoet and colleagues characterized them with regard to their reactivity toward citrullinated peptides [13]. Following the discovery of multiple protein candidates eligible for peptidyl arginine deiminase (PAD) citrullination, “RA citrullinome”, referring to the collection of hundreds of citrullinated proteins identified in the serum and synovial fluid of RA patients, was introduced to the field of RA research in recent decades [14,15,16,17,18]. With its superb diagnostic specificity, immunopathogenic relevance and excellent clinical correlations [19,20,21,22], ACPAs were included in the 2010 American College of Rheumatology (ACR)/European League Against Rheumatism (EULAR) RA classification criteria [23]. Alongside the classical RA autoAb RF, ACPA was also considered an important hallmark of the disease in the past decade.

In recent years, growing evidence has supported that proteins resulting from PTMs, other than citrullination, are also capable of triggering autoimmune responses important for the development of RA [24]. Indeed, autoantibodies against carbamylated protein (anti-CarP Ab), acetylated proteins, malondialdehyde, malondialdehyde-acetaldehyde and other posttranslated modified epitopes, such as anti-hinge antibodies, have attracted attention due to their clinical significance and their potential use in refining RA diagnosis and care [3,11,25,26,27]. Among them, anti-CarP Ab, described in 2011 [28], is perhaps one of the best studied anti-PTM autoantibodies.

The pathogenic roles of RA-associated autoAbs have been critically reviewed by us and other researchers [22,23,24,25,26,27,28,29]. In this review, we summarize the three most important RA-associated autoAbs, namely, RF, ACPA and anti-CarP Ab, regarding their distinct characteristics and clinical applications in the early diagnosis and prognosis prediction of RA. Moreover, with the advancement of treatment options, we also discuss the relevance of these autoAbs in choices of therapeutic regimens and clinical outcomes.

## 2. Characteristics of RF, ACPA and anti-CarP Ab

The function of an antibody is governed by the specific antigen it recognizes and the specific isotype determine by its crystallizable fragment (Fc) region. We first examine differences and similarities among RA-associated autoAbs for antibody characterization.

### 2.1. Rheumatoid Factors

Rheumatoid factors, despite their name, are not specific to RA. In fact, RFs are commonly produced during immunizations and secondary immune responses to infections to aid in pathogen removal [30,31,32]. Other rheumatic conditions, such as systemic lupus erythematosus, Sjogren’s disease and sarcoidosis, are also associated with the presence of RF [33]. Among healthy individuals, moreover, the presence of RF increases with age and exceeds 25% among elderly individuals over the age of 85 [34]. In contrast to the physiological RFs in healthy individuals, RFs isolated from RA patients have been reported to be relatively monoreactive, possessing higher affinity, harboring more somatic mutations, and utilizing more heterogeneous V-genes and more frequently become isotype switched [35]. Although a spectrum of immunoglobulins, including IgG, IgA, IgM and IgE, are all found among RFs, IgM, comprising the majority of RF isotypes in RA, is detected in 60–80% of RA patients, followed by IgA and IgG [36,37].

Interestingly, while it is generally believed that RF targets the CH2-CH3 Fc portion of IgGs [38], a recent study reported by Maibom-Thomsen et al. showed that RF does not interact with native IgGs in their soluble form when they are not bound to their corresponding antigens [39]. Through further mass spectrometry analysis, it was found that the binding of antibodies to pathogen surfaces likely induces a conformational change and exposes the RF epitopes within the Fc region (Figure 2) [39]. This novel finding provided an opportunity for a better understanding of the antibody structures and the functional physiology of antibodies in the human body.

### 2.2. Anti-Citrullinated Protein Antibodies

As depicted in Figure 2, citrullination is an enzymatic reaction mediated by PAD that converts arginine to citrulline. Keratin, fibrinogen, vimentin, fibronectin, α-enolase and 78-kDa glucose-regulated protein (GRP78) are well-known substrates for citrullination [14,15,16,17,18,40]. With the expanding collection of citrullinated proteins identified from the synovial fluid and serum of patients with RA, the term “RA citrullinome” has been introduced in recent decades [14,15,16,17,18]. However, despite its name, the spectrum of epitopes recognized by ACPAs is not limited to the “RA citrullinome” but also to peptides that undergo other PTMs, including carbamylation and acetylation [41,42]. Overlapping reactivity is commonly believed to explain the broad reactivity of ACPAs. Recently, Li and colleagues discovered that one-third of ACPAs recognize only monotargets, with limited overlapping reactivity [43]. Further evidence suggests that the majority of ACPAs in the sera of RA patients reacting to citrulline side chains have no functional role [44]. In fact, only ACPAs interacting or cross-reacting with citrulline and proximal amino acid side chains of articular proteins display arthritogenic capacity [44]. Additionally, the diversity and avidity of ACPAs toward citrullinated peptides have been noted to change and evolve over time. Kongpachith and colleagues found that somatic hypermutations accumulating during affinity maturation by clonally related B cells alter the antibody paratope to mediate “epitope spreading” and the polyreactivity of the ACPA response in patients with RA [45].

Similarly to the broad spectrum of RF isotypes, enrichment of IgA and IgG ACPA, particularly IgG1, occurs in the serum of RA patients and precedes the development of the disease [37,46,47,48]. Nevertheless, while glycosylation of the antigen-binding fragment (Fab) was estimated to be approximately 14% among global IgGs [49], more than 80% of citrulline-reactive B cells were found to contain glycosylation sites in their variable domains [50], and the variable domains of more than 90% of ACPAs indeed carried glycans [51]. This N-linked glycosylation in the Fab portion has been shown to critically predict the development of RA [52]. Glycosylation of the Fc fragment is also a unique feature of ACPAs [46,53]. Specifically, increasing core fucosylation and decreasing galactosylation and sialylation have been observed in the Fc fragment of ACPAs, directing molecular interactions and functions of ACPAs [46,53].

ACPAs, as compared to the classic RA antibody RF, have a superior diagnostic specificity and can be detected in approximately two-thirds of RA patients [19,20]. While 1–3% of healthy subjects may also test positive for ACPAs, their levels are usually in the lower range. Moreover, in comparison to the ACPAs isolated from the RA patients, those found in healthy individuals usually recognize a narrow spectrum of citrullinated antigens with altered avidities [43,54,55,56].

### 2.3. Anti-Carbamylated protein Antibodies

Anti-CarP Ab is another well-studied RA-related autoantibody. In contrast to citrullination, carbamylation is a nonenzymatic PTM of proteins that requires the presence of cyanate (Figure 2). Upon the binding of cyanate in the form of isocyanic acid to the ε-NH2 group side chain of lysine, homocitrulline is produced via the carbamylation reaction [57]. Nevertheless, under physiological conditions, the level of cyanate is generally too low to efficiently induce carbamylation. As urea is a source of cyanate, the level of cyanate in patients with elevated blood urea nitrogen and renal dysfunctions may be increased, promoting carbamylation [58]. Additionally, cyanate can be derived from the transformation of thiocyanate mediated by myeloperoxidase (MPO). Under chronic inflammatory conditions, MPOs may be released from neutrophils to enhance carbamylation [57].

Carbamylated alpha-1 anti-trypsin, fibrinogen, vimentin, alpha-enolase and GRP78 are some of the targets of anti-CarP Ab identified in the sera of patients with RA [57,59,60,61,62]. The avidity of ACPAs is considered low, and the even lower avidity of anti-CarP Abs suggests that despite proper class switching, no or little avidity maturation occurs at the time of the latter’s production [63,64]. Notably, anti-CarP Ab is found in nearly 45% of early RA patients positive for ACPAs, possibly because of close similarity between citrulline and homocitrulline [28,65,66]. Regardless, a large proportion of ACPA and anti-CarP Abs that interact only with citrullinated or carbamylated proteins can also be found alongside those that cross-react with one another in the serum of double-positive RA patients [42]. Through the investigation of a small RA cohort, Shi et al. reported a median percentage of noncross-reactive anti-CarP Ab as high as 70% [42]. Nonetheless, MC Lu discovered that while there was a positive correlation of ACPA with anti-GRP78 antibody in patients with RA. Similar correlation was not seen with anti-CarP GRP78 antibody [62]. Taken together, these findings suggest that anti-CarP Abs and ACPAs are not the same.

Specifically, the sensitivity of anti-CarP Ab is 18–26% and 27–46% prior to and after RA diagnosis, respectively, and its specificity is approximately 90% and above in RA [67]. Approximately 8–16% of ACPA-negative patients test positive for anti-CarP Ab [28,65]. Although anti-CarP Abs can also be found in healthy individuals and in other rheumatic conditions [68,69,70,71,72], the prevalence of anti-CarP Ab in RA patients is relatively high [71,73,74]. Similar to that of ACPA, isotypes of anti-CarP Ab display a broad spectrum. IgA, IgM and IgGs, including IgG1, IgG2, IgG3 and IgG4 anti-CarP Ab have all been discovered in patients with RA [73]. Specifically, anti-CarP IgG comprises all anti-CarP IgG subclasses, and IgA can be found in nearly 45% of RA patients, whereas anti-CarP IgM was detected in only 16% [73].

## 3. Risk Factors for the Production of RF, ACPA and Anti-CarP Ab

### 3.1. Genetic Risk Factors

Genetic background is an important factor influencing the development of RA. There is an estimated 40–50% familial risk among seropositive RA patients with strong risk noted in first-degree relatives [75]. In addition to the high correlation for RF reported among identical twins with RA [76], a number of studies have identified differences in genetic components between RA patients positive or negative for RF [77,78]. However, controversies were noted, as some studies demonstrated that RA patients, regardless of the presence of RF, may harbor similar human leukocyte antigen (HLA) susceptibility alleles [79,80].

A conserved amino acid sequence at positions 70 and 74 within the HLA-DRB1 molecule, namely, the “shared epitope (SE)”, is a leading genetic risk factor for ACPA-positive RA [81]. Furthermore, a dose effect of SE alleles on risk of RA among ACPA-positive RA patients was documented by Huizinga and others [82]. Utilizing paired samples of presymptomatic individuals, Kissel and others recently discovered that SE alleles are associated with ACPA Fab glycosylation in the predisease phase [83]. In addition to SE, the protein tyrosine phosphatase nonreceptor type 22 risk allele displays a synergistic action along with the SE alleles [84]. Furthermore, single-nucleotide polymorphisms and long noncoding RNAs are associated with the presence of ACPAs in RA patients [85,86]. Tumor necrosis factor (TNF) receptor-associated factor 1 C5 region, TNF-α-induced protein, CD40, C-C motif chemokine ligand 2, antisense noncoding RNA in the INK4 locus, and peptidyl arginine-deiminase 4 are some of the well-documented genetic factors linked to ACPA-seropositive RA [81].

In contrast, the genetic risk for anti-CarP Ab is less known. While HLA-DR3 is found more commonly in individuals with ACPA-negative RA than in the healthy population [87], HLA-DR3 is also associated with RA cases positive for anti-CarP Ab and negative for ACPA [88].

### 3.2. Environmental Risk Factors

#### 3.2.1. Tobacco Smoking

The significance of tobacco smoking as a risk factor for RA has been well accepted [89,90]. Interestingly, although smoking has been associated with all seropositive RA [54], it is known to be associated with the presence of RF, even in the absence of RA [91]. Large population studies and twin studies have described the association between smoking and ACPA positivity [92], with a dominant risk being in individuals positive for SE [92,93]. Aside from the inflammatory changes in the lungs subsequently causing activation of PADs [94], in the setting of SE, smoking confers a risk of high ACPA levels and suggests a distinct mechanism of ACPA production different from that of RF [91,95].

Moreover, smoking is suspected to promote carbamylation in patients with RA. While smoking has been demonstrated to promote MPO-mediated conversion of thiocyanate to cyanate [96], the level of anti-CarP Ab, however, was not significantly higher [68]. Perhaps critical factors (genetic or environmental exposures) other than carbamylation alone are required for the induction of anti-CarP Ab.

#### 3.2.2. Microbial Triggers

As RF is physiologically important to enhance immune complex clearance, to assist B cell uptake for antigen presentation and to facilitate complement fixation, an increased level of RF has been detected in individuals with chronic or indolent infection, including hepatitis B or C virus infection or infective endocarditis [97,98]. Nonetheless, the production of RF under such conditions typically ceases following resolution of the infection. Due to the action of bacterial pore-forming virulence and calcium ionophores in triggering calcium influx and generating nontolerized neocitrullinated epitopes [99,100,101], the role of periodontitis-causing bacteria and intestinal microbiota in ACPA-associated RA was recently summarized [102,103,104,105,106]. Although infection is likely to trigger the release of MPO from activated neutrophils during infection episodes, no direct evidence has linked acute infection to anti-CarP Ab-positive RA, and the risk of infection in autoantibody-mediated RA is still under investigation.

## 4. Predictive Values of RF, ACPA and Anti-CarP Ab in the Diagnosis of Presymptomatic RA Patients

Although not necessarily required for the development of RA, autoAbs can readily be found many years before the onset of symptomatic disease in the evolution model of RA [107,108,109]. During the presymptomatic phase of the disease, B cells undergo a series of selection, expansion, epitope spreading and antibody maturation processes [22,29], providing physicians an opportunity for early detection and prevention of the disease.

### 4.1. Rheumatoid Factor

The detection of RFs in the form of IgM, IgA, and IgG was found to predate the onset of RA by years across patients of various ethnicities [110,111,112]. Interestingly, their appearance in serum is sequential before the diagnosis of RA: IgM RF first, followed by IgA RF, and finally IgG RF [112,113]. In a Danish cohort study, healthy individuals with elevated RF levels had up to 26-fold higher long-term risk of RA and up to 32% 10-year absolute risk of developing RA [110]. Moreover, the positive predictive value (PPV) for RF ranges from 36–97%, with most values falling between 70% and 80%; the negative predictive value (NPV) is 69–95% [114].

### 4.2. Anti-Citrullinated Protein Antibodies

Similar to RF, ACPAs can be detected in serum samples up to 14 years before the onset of articular symptoms of RA and precede the presence of IgM RF by up to a decade [115]. Careful investigation of pre-RA blood donors and blood samples collected from patients before their diagnosis suggested that an increase in the level of ACAP can generally be found 1–3 years prior to the onset of symptoms [115,116]. In addition to the occurrence of ACPAs, the relevance of antibody levels and characteristics have been discussed in the prediction of RA development [117,118,119,120]. For example, while the PPV for the development of RA within a period of 2–6 years ranges from 30% to 70%, the best PPVs were those with higher antibody levels or those positive for both ACPA and RF [120,121]. In a study analyzing 260 IgM-RF-negative patients with early arthritis, the PPV of a positive ACPA test was 91.7% after a one-year follow-up [122].

In 2010, Van der Woude et al. discovered an excess amount of citrullinated epitopes recognizable by ACPA in the sera of pre-RA patients [123], reporting that the ACPAs isolated from those who were later confirmed to have RA recognized considerably more citrullinated targets [123]. Utilizing citrullinated peptide tetramers, a similar phenomenon was also reported recently by Kongpachith and colleagues [45]. Leaving a trail during the process of antibody maturation, the amount of ACPA epitope spreading was increased in the presymptomatic phase, predicting progression to RA [124,125]. Moreover, N-linked glycosylation in the Fab segment of ACPA is known to be critical in predicting RA development [52]. By studying a subset of first-degree relatives of indigenous North American RA patients, Hafkenscheid et al. discovered that extensive glycosylation of the IgG ACPA V domain predisposed individuals to the development of RA [52]. Altered glycosylation in the Fc portion as well as decreasing galactosylation and increasing fucosylation of serum ACPA IgG1 are also reported to precede the onset of RA [46].

### 4.3. Anti-Carbamylated Protein Antibodies

Similarly, anti-CarP Ab, which can be detected before the onset of RA, is also applicable in the setting of preclinical screening [25,126,127,128,129]. In fact, anti-CarP Ab can be detected more than a decade prior to the onset of RA, at approximately the same time as ACPA and before IgM RF can be detected [126,130]. Specifically, in comparison with those targeting carbamylated fibrinogen, the level of anti-CarP Ab screening against carbamylated fetal calf serum is more associated with future RA diagnosis according to Brink’s observation [129], who reported sensitivities of 13.9% and 42.2% for anti-CarP Ab in presymptomatic individuals and after the development of RA, respectively. Moreover, Pecani et al. recently reported a PPV of 88% and an NPV of 60% for anti-CarP Ab in RA patients [131].

### 4.4. Additive Values of the RF, ACPA and Anti-CarP Ab

Additive values of RF and ACPA in predicting the development of RA have been reported by Dahlqvist et al. and have been reproducibly confirmed [114,132]. In comparison to controls, the specificity for future development of classifiable RA with the combination of both ACPA and RF of any isotype can reach as high as 99% (Table 1) [112].

In 2015, Shi et al. analyzed sera derived from patients in the Leiden Early Arthritis Clinic cohort and reported a sensitivity and specificity of 44% and 89% for anti-CarP Ab in RA, respectively. Interestingly, these researchers observed that triple positivity for RF, ACPAs and anti-CarP Ab was almost exclusively present in RA and not found in other forms of arthritis [133]. However, investigating 1062 patients with early arthritis and following them up for two years, Regueiro and colleagues analyzed these three autoAbs and discovered that the association of anti-CarP Ab with RA was notably weaker than that of ACPA and RF [134]. In addition, they found only a mild increase in sensitivity when combining anti-CarP Ab into the diagnosis of RA compared with the current 2010 RA guidelines of counting ACPA and RF [134]. Specifically, the inclusion of anti-CarP Ab in the combined positivity criterion, namely, ACPA or RF or anti-CarP Ab, only resulted in an additional increase of 2.2% in sensitivity, but with a cost of 8.1% loss in specificity over the existing ACPA or RF positivity criterion [134]. Recently, Verheul et al. performed a meta-analysis utilizing 12 relevant articles and discovered that triple positivity for anti-CarP Abs, ACPAs and RF contributes a higher specificity (98–100%) but lower sensitivity (11–39%) in RA [27]. Furthermore, Regueiro and colleagues evaluated the potential of combining the three antibodies for improving current RA classification among patients with early arthritis [11]. They reported that the presence of the antibodies together favors predictive characteristics for RA, with a PPV of 96.1%, which is better than the classification criterion currently available (PPV  =  88.8%) [11]. Due to the remaining controversy regarding whether it is beneficial to include anti-CarP Abs, debate over the value of combining the three autoAbs is ongoing [11,26].

## 5. Monitoring RA Disease Activity and Progression with RF, ACPA and Anti-CarP Ab

### 5.1. Rheumatoid Factor

Before the introduction of other autoAbs, RF alone, particularly IgM and IgA isotypes at high titers, had been a well-known factor predicting more aggressive and erosive joint disease with a higher prevalence of extraarticular manifestations [98,135]. Moreover, the presence of IgM RF in patient sera renders clinical remission less achievable. In fact, a recent study reported that RF is associated with early methotrexate (MTX) failure due to inefficacy in patients with early RA [136].

Some studies have shown that the level of RF can decrease following proper immune suppressant treatment [137,138]. In a review by Ingegnoli et al., a progressive decrease in the level of RF was reported to parallel the decrease in disease activity noted in RA patients who underwent treatments, including conventional disease-modifying anti-rheumatic drugs (DMARDs) and various different biologics, including infliximab, etanercept, adalimumab, rituximab, and abatacept or tocilizumab [98]. However, there are conflicting results when utilizing RF to predict the response to therapeutic regimens and limit its clinical application [137,138]. For example, high levels of serum RF are reasonable predictors for a better response to B cell-depleting therapy. Indeed, most patients positive for RF exhibit a moderately better response to rituximab than those who are negative [139]. Nevertheless, the response to anti-TNF varies significantly. While some studies claimed that RF positivity before therapy predicted a favorable response, others reported opposite findings [98]. Evidence gathered from systematic reviews and meta-analyses of clinical trials and observational studies, including 14 studies of rituximab treatment and 6 studies of tocilizumab treatment, revealed that RF positivity at initiation predicts a better clinical response [140]. There is, however, no association between RF and response to abatacept [140].

### 5.2. Anti-Citrullinated Protein Antibodies

While the clinical significance and the immunopathogenesis of RF and ACPA were well elucidated, no tailored treatments are preferentially recommended for seropositive RA patients or those with high ACPA specifically [141]. According to the long-term study of the BeSt strategy trial, despite a greater extent of radiological articular damage, patients positive for ACPA achieved a reduction in disease activity that was identical to those who were negative [142]. Follow-up studies by Dekkers and Jonsson suggested that short-term remission or different EULAR responses in newly diagnosed RA were not affected by either ACPA or RF positivity among patients who underwent MTX monotherapy [143,144]. Although the presence of ACPA does not seem to add much to the prediction of a treatment response to MTX, reports have suggested some differences in the magnitude of the response influenced by treatment dosing. Indeed, Wevers-de Boer and Crepaldi suggested that higher starting doses of MTX were more effective than lower doses in treating seropositive than seronegative RA patients [145,146]. Furthermore, Seegobin et al. further proposed that triple therapy with MTX, cyclosporine and prednisolone is more effective in reducing disease activity solely in ACPA-positive early RA patients [147].

A recent study utilizing head-to-head comparisons in a real-world setting was conducted to compare the impact of seropositivity (ACPA and RF positive) on drug discontinuation and the effectiveness of various biologics in patients with RA [148]. Analyzing a pooled of 16 observational RA registries, including 27,583 eligible patients, Courvoisier et al. found seropositivity to be associated with longer drug retention and decreased disease activity for rituximab and abatacept. In addition, there were slight associations between seropositivity and effectiveness for tocilizumab but not TNF inhibitors [148]. A similar observation was also made by Bugatti, who examined each study in detail [149]. Although rituximab and abatacept have been shown to partially reduce the level of ACPA in RA patients in close association with a reduction of disease activity [150,151], Cambridge et al. discovered that there was no increase in the level of ACPA prior to or during relapse following initial response [152]. Hence, the role of ACPA as a biomarker for disease activity hence is still questionable [29,153].

Following the introduction of Janus kinase inhibitors to the treatment of RA, interest in how seropositivity predicts its treatment response has emerged. Recently, Bird et al. analyzed 3061 patients pooled from five RA cohorts and discovered that those who were positive for both ACAP and RF were more likely to achieve clinical response with tofacitinib than those who were negative for both autoAbs [154]. Additionally, patients who were positive for ACPA, regardless of RF status, were more likely to achieve disease remission or low disease activity than seronegative patients when treated with tofacitinib 10 mg twice daily [154]. More studies are required to draw definitive conclusions.

### 5.3. Anti-Carbamylated Protein Antibodies

Anti-CarP Ab is detected in up to 45% of RA patients, though its presence was found in only 16–30% of those negative for ACPA [28]. Despite conflicting results, the majority of studies agree that the presence of anti-CarP Ab is associated with a higher degree of disease activity and significantly more disability over time in patients with RA [74,128,155]. Recently, more severe radiological progression was reported in patients positive for anti-CarP IgG, specifically among those negative for ACPA. This indicates a role of anti-CarP Ab as a unique and relevant serological marker for RA patients negative for ACPA [28,156].

Kumar et al. recently reported a correlation between anti-CarP positivity at baseline and a reduction in disease activity within the first six months of treatment among 60 RA patients treated with abatacept [157]. However, the predictive role of anti-CarP Ab in response to RA treatment in other treatments has not been extensively investigated. More research appears to be needed to determine the usefulness of anti-CarP Ab for monitoring RA disease activity and therapeutic response.

## 6. Summary and Future Prospects

Studies on RA-associated autoAbs not only facilitate our understanding of disease immunopathogenesis but also, as summarized in Table 1, have allowed these molecules to be adopted clinically as biomarkers for disease diagnosis and outcome prediction and for directing medication choice. In fact, because RA-associated autoAbs appear before the onset of disease, multiple living and pharmacological strategies have been proposed and tested to delay or prevent RA development [158,159,160]. Finckh et al. specifically highlighted cases associated with genetic factors, such as SE, and patients with systemic autoimmunity associated with ACPA-positive RA for arthritis prevention [161]. In the ADJUST trial, treatment with abatacept was able to postpone the progression of arthritis by modulating T-cell responses in some early-stage ACPA-positive RA patients [162]. The results also reveal that the progression of undifferentiated arthritis and very early RA can be therapeutically manipulated in a proportion of patients [162]. Moreover, the ongoing STAPRA trial, StopRA study and APIPPRA trial aim to prevent progression of the disease among patients with high titers of ACPA or ACPA and RF IgM utilizing atorvastatin, hydroxyquinone and abatacept before the onset of RA [159]. Discontinuation of smoking and eradication of mucosal infection are some modifiable risk factors worth pursuing [163,164,165]. With the introduction of RF, ACPA, anti-CarP Ab and upcoming novel autoAbs, these clinically available biomarkers are of value in assisting in the diagnosis and treatment of patients with RA.

## Figures and Tables

**Figure 1 ijms-22-00686-f001:**
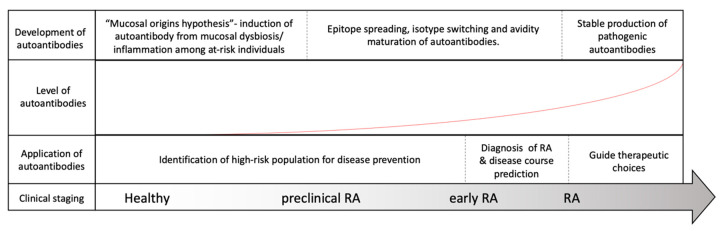
Evolution of rheumatoid arthritis (RA) and its correlation with RA-associated autoantibodies. Among at-risk populations, antigen exposure from the mucosal area helps form T-cells and assists in the induction of RA-associated autoantibodies many years before the onset of RA. The development of RA, however, requires further autoantibody maturation, including epitope spreading, isotype switching, avidity maturation and antibody glycosylation, before they are pathogenically transformed. As our understanding of autoantibodies advances, the application of the autoantibodies may be utilized to assist in RA diagnosis, to predict disease course, to guide treatment and possibly to prevent or delay RA development before its onset.

**Figure 2 ijms-22-00686-f002:**
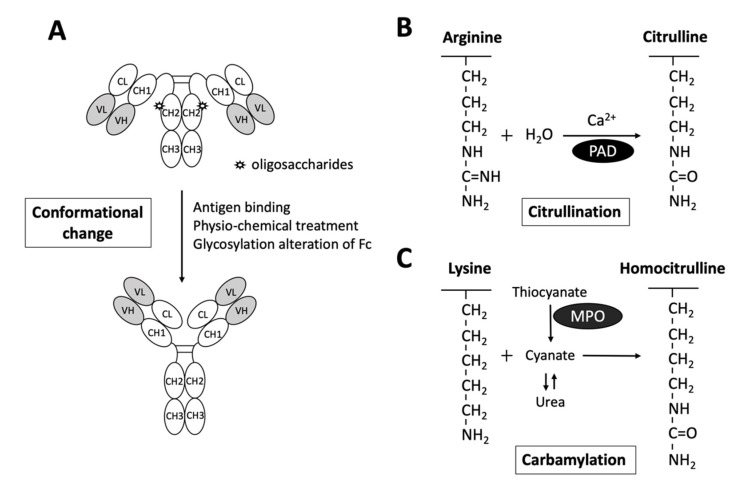
Schematic diagram illustrating the exposure of RF epitopes and posttranslational modification of peptide targets for autoantibody interactions in rheumatoid arthritis. (**A**) The epitopes of RFs reside in the Fc portion of IgG mostly in the CH2/CH3 or the CH3/CH3 groove. Recent evidence has suggested that conformational changes via altered Fc glycosylation, physicochemical treatment, antigen binding, or physical adsorption onto a hydrophobic surface promote the exposure of RF epitopes on circulating IgGs. (**B**) Citrullinated peptides are generated by the enzymatic activity of peptidylarginine deiminase (PAD), converting arginine to citrulline in the presence of calcium. (**C**) Carbamylation is the conversion of a lysine to homocitrulline upon the binding of cyanate in the form of isocyanic acid to lysine, without enzymatic catabolism. Urea and thiocyanate catabolized by myeloperoxidase (MPO) are important sources of cyanate.

**Table 1 ijms-22-00686-t001:** Comparison of the characteristics and clinical significances between rheumatoid factors, anti-citrullinated protein antibodies and anti-carbamylated protein antibodies.

	Rheumatoid Factor	Anti-Citrullinated Protein Antibodies	Anti-Carbamylated Protein Antibodies
**Year of ** **Discovery**	1940s	1964	2011
**Characteristics of the Autoantibody**			
**Antigenic ** **Targets**	- target the antigenic epitopes within the Fc region (CH2 and CH3) of IgG - conformation changes may be required for epitope exposure	- react with a variety of citrullinated antigens - partially cross-react with peptides undergone carbamylation and acetylation	- react with a variety of carbamylated antigens, homocitrulline - partially cross-react with peptides undergone citrullination
**Isotypes**	- IgM > IgG > IgA	- mainly IgG and IgA	- mainly IgG and IgA
**Characteristics**	- limited N-glycosylation - limited somatic hypermutations - limited class switching	- extensive N-glycosylation - extensive somatic hypermutations - extensive class switching	- extensive class switching - limited avidity maturation
**Clinical Significance**			
**Role in RA ** **Diagnosis**	- included in the 1987 ACR classification criteria for RA - included in the 2010 ACR/EULAR RA classification criteria	- included in the 2010 ACR/EULAR RA classification criteria	- not included in any RA classification criteria - potentially beneficial for those negative of RF and ACPA
**Sensitivity for RA**	- RF: 41–66% for early RA and 62–87% for RA	- ACPA: 41–66% for early RA and 41–77% for RA - double positive: 33–57% for RA	- anti-CarP Ab: 18–26% prior to and 27–46% after RA diagnosis - triple positivity: 11–39%
**Specificity for RA**	- RF: 43–96% for RA	- ACPA: 88–98% for RA - double positive: 91–99%	- anti-CarP Ab: 93–97% for RA - triple positivity: 98–100%
**Other Conditions with ** **Increased Antibody Titer**	- other autoimmune diseases - chronic infections/inflammations - aging - smoking	- mucosal dysbiosis - tobacco smoking - other autoimmune diseases	- renal diseases - chronic inflammations - cardiovascular disease - other autoimmune diseases
**Clinical Presentations**	- more extraarticular manifestations - aggressive and erosive joint disease	- more severe structural damage, radiographic progression and poorer response to therapy	- more severe joint damage in those negative for ACPA
**Treatment ** **Response**	- the level of RF parallels the decrease of disease activity - RF positivity possibly predicts better clinical response in those receiving rituximab and tocilizumab	- aggressive treatment with higher MTX dosing or triple therapy may be beneficial for ACPA RA - ACPA positivity cases possibly response to tocilizumab, rituximab, abatacept and tofacitinib better than those negative of ACPA	- Abatacept seem beneficial for those positive for anti-CarP Ab

RA, rheumatoid arthritis; RF, rheumatoid factors; ACPA, Anti-citrullinated protein antibody; anti-CarP Ab, Anti-carbamylated protein antibody; ACR, American College of Rheumatology; EULAR, European League Against Rheumatism; MTX, Methotrexate; Fc-crystallizable fragment.

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
