# Peer review of "From Rheumatoid Factor to Anti-Citrullinated Protein Antibodies and Anti-Carbamylated Protein Antibodies for Diagnosis and Prognosis Prediction in Patients with Rheumatoid Arthritis"

_ijms, 2021, doi:10.3390/ijms22020686_

Round 1

Reviewer 1 Report

Dear authors,

This is an interesting review article described : From rheumatoid factor to anti citrullinated protein, antibodies and anti-carbamylated protein antibodies, for diagnosis and prognosis prediction in patients with rheumatoid arthritis. It is a well-done review. It’ll be better if the authors can add Dr. Henry Kunkel for his great efforts in the study of RF in the early stage and Dr. MC Lu for his outstanding finding in the ACPA protein.

Author Response

We thank the reviewer very precious comments on our manuscript. The suggestion has been well-received and the changes are made accordingly. In the revised manuscript, four more references and the related information have already been added and incorporated into the text.

References

7. Franklin EC, Holman HR, Muller-Eberhard HJ, Kunkel HG. An unusual protein component of high molecular weight in the serum of certain patients with rheumatoid arthritis. J Exp Med. 1957;105(5):425-438.

8. Edelman GM, Kunkel HG, Franklin EC. Interaction of the rheumatoid factor with antigen-antibody complexes and aggregated gamma globulin. J Exp Med. 1958;108(1):105-120.

40. Lu MC, Lai NS, Yu HC, Huang HB, Hsieh SC, Yu CL. Anti-citrullinated protein antibodies bind surface-expressed citrullinated Grp78 on monocyte/macrophages and stimulate tumor necrosis factor alpha production. Arthritis Rheum. 2010;62(5):1213-1223.

62. Yu HC, Lai PH, Lai NS, Huang HB, Koo M, Lu MC. Increased Serum Levels of Anti-Carbamylated 78-kDa Glucose-Regulated Protein Antibody in Patients with Rheumatoid Arthritis. Int J Mol Sci. 2016;17(9).

Reviewer 2 Report

This is an interesting review about three types of autoantibodies, which are very frequently detected in the sera of patients with rheumatoid arthritis (RA): rheumatoid factor, anti-citrullinated, and anti-carbamylated autoantibodies. The authors focus their attention in the potential use of these antibodies for the diagnosis and prognosis of RA. I do not have specific questions to the authors besides the fact that a summary abstract may be helpful to non-specialists to keep their attention in the manuscript.

Author Response

We thank the reviewer very careful reading of our manuscript. In response to the reviewer's comment, we rewrite the abstract to meet the reviewer's concern. The changes are highlighted in red color in abstract (after modification, the word counts were still kept to meet the limitation of the journal's requirement).